# The effect of mental health on sleep quality of front-line medical staff during the COVID-19 outbreak in China: A cross-sectional study

Leiyu Yue[1], Rui Zhao[2], Qingqing Xiao[1], Yu Zhuo[1], Jianying Yu[1], Xiandong Meng[1]*

1 West China School of Nursing, Sichuan University/Department of Mental Health Centre, West China Hospital, Sichuan University, Chengdu, PR China, 2 Geriatric Psychiatric Ward 2, The Fourth People's Hospital of Chengdu, Chengdu, PR China

☯ These authors contributed equally to this work.
* wchmhcmxd@163.com

## Abstract

### Background

The 2019 coronavirus disease (COVID-19) pandemic is a public health emergency of international concern and poses a challenge to the mental health and sleep quality of front-line medical staff (FMS). The aim of this study was to investigate the sleep quality of FMS during the COVID-19 outbreak in China and analyze the relationship between mental health and sleep quality of FMS.

### Methods

From February 24, 2020 to March 22, 2020, a cross-sectional study was performed with 543 FMS from a medical center in Western China. A self-reported questionnaire was used to collect data anonymously. The following tests were used: The Self-Rating Anxiety Scale (SAS) for symptoms of anxiety, the Beck Depression Inventory (BDI) for depressive symptoms, and the Pittsburgh Sleep Quality Index (PSQI) for sleep quality assessment.

### Results

Of the 543 FMS, 216 (39.8%) were classified as subjects with poor sleep quality. Anxiety (P<0.001), depression (P<0.001), and the prevalence of those divorced or widowed (P<0.05) were more common in FMS with poor sleep quality than in participants with good sleep quality. The FMS exhibiting co-occurrence of anxiety and depression were associated with worse scores on sleep quality than those medical staff in the other three groups/categories. The difference in sleep quality between the FMS with only depression and the FMS experiencing co-occurrence of anxiety and depression was statistically significant (P<0.05). However, there was no significant difference in sleep quality between the FMS experiencing only anxiety and the FMS with co-occurrence of anxiety and depression (P > 0.05).

**Data Availability Statement:** All relevant data are within the paper and its S1 File.

**Funding:** This research was received grant from funding 2019 novel coronavirus disease

technology research project of West China Hospital of Sichuan University (NO: HX-2019-nCov-034). The funders had no role in study design, data collection and analysis, decision to publish, or preparation of the manuscript.

## Conclusions

During the COVID-19 pandemic, there was a noteworthy increase in the prevalence of negative emotions and sentiments among the medical staff, along with poor overall sleep quality. We anticipate that this study can stimulate more research into the mental state of FMS during outbreaks and other public health emergencies. In addition, particular attention must be paid to enhance the sleep quality of FMS, along with better planning and support for FMS who are continuously exposed to the existing viral epidemic by virtue of the nature of their profession.

## Introduction

Coronavirus disease 2019 (COVID-19), an infectious respiratory disease caused by a novel coronavirus strain, known as severe acute respiratory syndrome coronavirus 2 (SARS-CoV-2), was first identified in December 2019 in Wuhan City, in central China and spread rapidly to the rest of the world, including Europe and the United States [1]. It has a high transmission rate and can be transmitted via close human-to-human contact [2, 3]. The World Health Organization (WHO) has declared COVID-19 a Public Health Emergency of International Concern as of 1 February 2020 [4]. Data as received by WHO from national authorities, as of 21 February 2021, there have been 110.7 million confirmed cases of COVID-19, including 2.4 million deaths since the start of the pandemic [5].

As the focal point of epidemic prevention and control, hospitals are the principal settings of confirmed or suspected cases of COVID-19, making them the most susceptible sites for new infections. In the wake of the current viral pandemic, the front-line medical staff (FMS) have indubitably been the most impacted groups, with increased workload involving diagnosis and treatment of new infections, elevated stress levels, reduced or overwhelmed health system capacity, and increased risk of infections [6]. These constant stressors may adversely impact sleep quality and mental health of FMS. In a meta-analysis showed that anxiety was assessed in 12 studies, with a pooled prevalence of 23.2% and depression in 10 studies, with a prevalence rate of 22.8% during the COVID-19 pandemic [7].

It has long been known that medical staff often suffer from sleep disorders and low sleep quality, due primarily to work-related stressors, sleep deprivation and shift work [8]. The COVID-19 outbreak in China remains to be a serious challenge for FMS. These professionals, by virtue of their continuous and intimate association with patients, are not only under high risk of getting infected themselves, but they also suffer from high mental stress, which may lead to sleep disturbances [9]. Qiu et al. [10] found that 39.2% of Chinese medical workers suffered from sleep disorders. However, 51.7% of front-line medical staff had sleep disorders under COVID-19 outbreak in China [9]. Sleep disorders not only increase the risk of infection among medical staff, but also impair their work performance during the pandemic [11, 12], and has a negative impact on health including increased risk of stroke, obesity, diabetes, cancer, osteoporosis, and cardiovascular disease [13]. More alarmingly, individuals experiencing persistent and progressive decline in sleep quality are more likely to develop mental illnesses (such as anxiety and depression) [14] and exhibit suicidal behaviors [15].

Early prevention, proper recognition/diagnosis and treatment of anxiety and depression, along with strategies aimed at improving sleep quality are especially crucial during extraordinarily stressful times, such as during the COVID-19 pandemic, because these strategies may

significantly prevent the re-emergence of sleep disorders and mental illness [16]. Maintaining a good sleep quality and mental health can not only help FMS better treat patients, but also help them maintain optimum immune function and prevent infections [17]. Therefore, the purpose of this study was to investigate the mental state and sleep quality of the FMS during the COVID-19 pandemic, to explore the risk factors underlying poor sleep quality, analyze the relationship between mental state and sleep quality, to provide a scientific basis for the prevention and control of mental disorders, and, finally, to suggest strategies aimed at enhancement of sleep quality of medical staff. Further, this study can serve as a primary reference and information guide for hospitals in other countries to help them maintain the mental and physical health of their medical staff as they continue to deal with the possible resurgence of the COVID-19 global pandemic.

## Methods

### Ethical approval

The study was approved by the Biomedical Research Ethics Committee, West China Hospital of Sichuan University (Approval number: 20200220).

### Study design

An observational and cross-sectional clinical study was conducted that included the use of self-reported questionnaires. The questionnaire was built on a professional questionnaire survey network platform called "Wenjuan Xing" (www.wjx.cn) and then was shared on social media WeChat. While constructing the online questionnaire, the integrity check function of the platform was used, meaning the questionnaire could not be submitted unless all questions were answered. All questionnaires were completed anonymously by FMS.

### Study participants

We contacted department heads in each department and invited them to forward our questionnaire to their WeChat group of staff to recruit participants. This study included 543 FMS from a medical center in Western China who regularly treated or were in contact with patients infected with COVID-19, during a period spanning from February, 2020 to March, 2020 by convenience sampling. The study participants included doctors, nurses and technicians who worked in high-risk Covid-19 clinical departments, laboratories, and administrative departments. The inclusion criteria were as follows: (1) regular employees, (2) worked at their posts during the survey, (3) WeChat users. All study participants willingly volunteered to participate in the study.

### Sample size

Sample size was determined using the formula:

$$N = \frac{z^2 \times p(1-p)}{e^2}$$

where 'z' is 1.96 at 95% confidence interval, 'e' is margin of error at 5% and 'p' is prevalence rate of 40% from a recent study done in China [10]. According to the formula, N = 369, considering the non-response rate of 20%, at least 443 sample size are needed.

## Measures and instruments

The online questionnaire had four sections: sociodemographic, depression symptoms, anxiety symptoms and sleep quality were required. After a brief written informed consent at the beginning of the survey, the questionnaire was answered. Sociodemographic data including age, gender, education, marital status, living with family members or not, employee type and seniority, were also required. Levels of anxiety, levels of depression, and sleep quality were measured using validated clinical questionnaires and scoring systems.

**The Self-Rating Anxiety Scale (SAS).** The Self-Rating Anxiety Scale (SAS) was compiled by William W. K. Zung in 1971 [18]. The SAS was used to measure the levels of anxiety of the medical staff, which contained 20 items consisting of four grades, with questions based on feelings of anxiety and mood in the previous seven days. An aggregate score of 20 was then multiplied by 1.25, with higher scores indicating more severe levels of anxiety. The demarcation value of SAS standard deviation is 50 points, with SAS $\leq$ 50 points judged as "no anxiety state", and > 50 points considered as "presence of an anxiety state". The Cronbach's alpha (Tau-equivalent reliability), as a measure of internal consistency for the use of SAS, was 0.821 [19].

**Beck Depression Inventory (BDI).** The BDI was compiled by the clinical psychologist, Aaron T. Beck in 1961 [20], which we employed to measure the levels of depressive mood of the medical staff. Although the BDI contains a 21-question self-report inventory in its original form, a 13-item abbreviated scale was developed in the Early Clinical Drug Evaluation Program and is widely used in research [21]. This refined/reformatted BDI contains 13-question self-rated inventory on a scale of 0–3 to give score of 0–39. The demarcation value of BDI standard deviation is 4 points, BDI $\leq$ 4 points is judged as "no depression", and BDI > 4 points is considered as "depression". The Cronbach's alpha, for internal consistency for the use of BDI, was 0.8847 [22].

**The Pittsburgh Sleep Quality Index (PSQI).** The Pittsburgh Sleep Quality Index (PSQI), developed by D. J. Buysse [23], was used to measure sleep quality using a 19-item scale, containing seven items that included sleep quality, sleep duration, sleep latency, habitual sleep efficiency, sleep disturbance, any use of sleeping medications, and daytime dysfunction over the last month. The seven-component scores are added together to get a global PSQI score. For descriptive purposes, participants with scores below 5 points were considered to have good sleep quality, whereas, participants with scores higher than 5 points had poor sleep quality. The Cronbach's alpha, as a measure of internal consistency for the use of the PSQI, was 0.811 [24].

## Statistical analysis

All data were analyzed using IBM SPSS version 24.0. Measurement data conforming to normal distribution were presented as the means and standard deviation of the mean. Categorical variables were expressed as absolute values and percentages. Measurements between groups and within groups were analyzed using the t-test, Chi-squared test with count data, grade data, using Kruskal-Wallis test analysis. Multivariate logistic regression analysis was performed on the variables that were significant in univariate analysis. The binary logistic regression analyses were used to estimate the odds ratio for each independent variable, to assess which of the factors associated with poor sleep quality. We used ANOVA to compare the differences of PSQI scores among the following groups: "Anxiety only", "depression only", "co-occurrence of anxiety and depression", "neither anxiety nor depression". Bonferroni's multiple comparison test was conducted to examine which two means were different. All data analyzed was set at a statistically significant level of p<0.05.

## Results

### Demographic data of the subjects

A total of 546 FMS completed the questionnaire survey, which included three participants were disagreeing to use their answers for study due to their answers are worthless(n = 2), worried about expose their privacy(n = 1). Hence,543 effective questionnaires, with an effective rate of 99.4%. Most of the participants were women, accounting for 94.3% of the total. In addition, 84.4% of the study subjects were nurses (Table 1).

**Analysis of factors affecting sleep quality during COVID-19 outbreak.** Among the 543 FMS, 43 (7.9%) were in an anxiety state (SAS>50 points) and 103 (18.9%) had depression (BDI > 4 points). Of the 543 respondents, 216 (39.8%) were classified as subjects with poor

**Table 1. Demographic of front-line medical staff (N = 543).**

| Variable | | Frequency | Percent (%) |
|---|---|---|---|
| **Gender** | Male | 31 | 5.7 |
| | Female | 512 | 94.3 |
| **Age (years)** | 18~34 | 324 | 59.7 |
| | 35~44 | 130 | 23.9 |
| | 45~54 | 77 | 14.2 |
| | 55~64 | 12 | 2.2 |
| **Marital status** | Married | 394 | 72.6 |
| | Unmarried | 137 | 25.2 |
| | Divorced/widowed | 12 | 2.2 |
| **Education** | College degree or below | 505 | 93.0 |
| | Bachelor's degree | 28 | 5.2 |
| | Master's degree or above | 10 | 1.8 |
| **Living condition** | Family cohabitation | 362 | 66.7 |
| | Live alone | 181 | 33.6 |
| **Department** | Clinical departments | 427 | 78.6 |
| | Executive branch | 23 | 4.3 |
| | Logistics department | 93 | 17.1 |
| **Profession** | Doctor | 43 | 7.9 |
| | Nurse | 458 | 84.4 |
| | Technician | 42 | 7.7 |
| **Working experience** | Mean ± SD | 12.0±9.6 | |
| **BDI score** | ≤4 | 440 | 81.1 |
| | >4 | 103 | 18.9 |
| | Mean ± SD | 2.4±3.89 | |
| **SAS score** | ≤50 | 500 | 92.1 |
| | >50 | 43 | 7.9 |
| | Mean ± SD | 37.1±8.76 | |
| **PSQI score** | ≤5 | 327 | 60.2 |
| | >5 | 216 | 39.8 |
| | Mean ± SD | 5.2±3.24 | |

BDI = Beck Depression Inventory.

SAS = Self-Rating Anxiety Scale.

PSQI = Pittsburgh Sleep Quality Index.

SD = Standard deviation.

sleep quality; anxiety [odds ratio (OR), 4.7; 95% confidence interval (CI), 2.0–11.2], depression (OR, 4.9; 95% CI, 2.9–8.3), and the proportion of those who had divorced/or were widowed [OR, 6.1;95% CI, 1.1–32.7] were more common in FMS with poor sleep quality than in those with good sleep quality (Table 2).

**Table 2. Analysis of factors affecting sleep quality [number (percentage %)].**

| Variables | N = 543 | Sleep Quality | | Univariate analysis | | Multivariate analysis | |
|---|---|---|---|---|---|---|---|
| | | Good (≤5) | Poor (> 5) | $t/\chi^2$ | p-value | OR (95%CI) | p-value |
| | | (n = 327) | (n = 216) | | | | |
| **Gender** | | | | | | | |
| Male | 31 | 21(6.4%) | 10(4.6%) | 0.766 | 0.378 | | |
| Female | 512 | 306(93.6%) | 206(95.4%) | | | | |
| **Age (years)** | | | | | | | |
| 18~34 | 324 | 205(62.7%) | 119(55.0%) | 6.498 | 0.09 | | |
| 35~44 | 130 | 75(22.9%) | 55(25.5%) | | | | |
| 45~54 | 77 | 38(11.6%) | 39(18.1%) | | | | |
| 55~64 | 12 | 9(2.8%) | 3(1.4%) | | | | |
| **Marital status** | | | | | | | |
| Married | 394 | 228(69.7%) | 166(76.9%) | 16.817*** | < 0.001 | 1.00 [Reference] | |
| Unmarried | 137 | 97(29.7%) | 40(18.5%) | | | 0.5(0.3–1.0) | 0.092 |
| Divorced or widowed | 12 | 2(0.6%) | 10(4.6%) | | | 6.1(1.1–32.7) * | 0.032 |
| **Education** | | | | | | | |
| College degree or below | 505 | 305(93.3%) | 200(92.6%) | 0.117 | 0.943 | | |
| Bachelor's degree | 28 | 16(4.9%) | 12(5.6%) | | | | |
| Master's degree or above | 10 | 6(1.8%) | 4(1.9%) | | | | |
| **Living condition** | | | | | | | |
| Family cohabitation | 362 | 212(64.8%) | 150(69.4%) | 1.245 | 0.264 | | |
| Live alone | 181 | 115(35.2%) | 66(30.6%) | | | | |
| **Department** | | | | | | | |
| Clinical departments | 427 | 250(76.5%) | 177(81.9%) | 2.699 | 0.259 | | |
| Executive branch | 23 | 14(4.3%) | 9(4.2%) | | | | |
| Logistics department | 93 | 63(19.3%) | 30(13.9%) | | | | |
| **Profession** | | | | | | | |
| Doctor | 43 | 24(7.3%) | 19(8.8%) | 0.401 | 0.818 | | |
| Nurse | 458 | 278(85.0%) | 180(83.3%) | | | | |
| Technician | 42 | 25(7.6%) | 17(7.9%) | | | | |
| **Working experience** | | 11.1±9.46 | 13.3±9.73 | -2.753** | 0.006 | 1.0(0.9–1.0) | 0.243 |
| **Anxiety** | | | | | | | |
| Yes | 43 | 9(2.8%) | 34(15.7%) | 30.09*** | < 0.001 | 4.7(2.0–11.2) *** | <0.001 |
| No | 500 | 318(97.2) | 182(84.3%) | | | 1.00 [Reference] | |
| **Depression** | | | | | | | |
| Yes | 103 | 27(8.3%) | 76(35.2%) | 61.366*** | < 0.001 | 4.9(2.9–8.3) *** | <0.001 |
| No | 440 | 300(91.7%) | 140(64.8%) | | | 1.00 [Reference] | |

OR = Odds Ratio.

CI = Confidence Interval.

*0.05 > p-value > = 0.01.

**0.01 > p-value > = 0.001.

***p-value < 0.001.

**Comparison of the total score and subscale scores for sleep quality of FMS with different demographic characteristics and mental states during COVID-19 outbreak.** The total score for sleep quality and other scores for its seven subscales in FMS with anxiety or depression were higher than those without anxiety or depression. The score for overall sleep quality, the amount of sleep and sleep efficiency of divorced/widowed FMS were higher than the other FMS. In addition, the study also found that in terms of sleep latency scores, the values were higher for female FMS than that of males. The sleep latency scores of the FMS living alone ware higher than that of those living with their families. The hypnotic drug score of FMS aged 55–64 years was higher than that of other age groups (P < 0.05) (Table 3).

**The effect of mental health on sleep quality during COVID-19 outbreak.** A total of 543 FMS were divided into four groups, according to whether they experienced anxiety or depression. There were 16 (2.9%) with "anxiety only" (Group 1), 76 (14.0%) with "depression only" (Group 2), 27 (5.0%) with "co-occurrence of anxiety and depression" (Group 3), and 424 (78.1%) with "neither anxiety nor depression" (Group 4). The results indicated that the PSQI score of the medical staff with co-occurrence of anxiety and depression was the highest and their sleep quality was the worst (Table 4). Multiple comparisons showed that there was a significant difference between Group 4 and the other three groups (group 1, 2, and 3) (P<0.05), suggesting that the sleep quality of FMS with only anxiety, only depression, and co-occurrence of anxiety and depression was worse than those with neither anxiety nor depression. The difference in sleep quality between Group 2 and Group 3 was statistically significant (P <0.05), indicating that the sleep quality of FMS with co-occurrence of anxiety and depression was worse than those with depression alone. However, no significant differences between Group 1 and Group 3 were observed (P > 0.05) (Table 5).

## Discussion

Previous studies have suggested that medical workers are particularly vulnerable to sleep disorders even during times of relative tranquility [25]. In addition, the rapid spread of the viral outbreak, inadequate early-stage testing/screening, and lack of targeted anti-viral treatments toward COVID-19 all constituted a pressing challenge for the FMS in numerous countries, exerting great psychological pressure on them. Therefore, the purpose of this study was to investigate the mental state and sleep quality of FMS during the COVID-19 outbreak in a hospital setting providing primary care and screen for COVID-19. Our results suggest that 7.9% FMS had anxiety and 18.9% had depression are lower than previous studies, reported in Wuhan during the same period. Potential differences, however, could be explained on the basis of the extremely high infectious potential rate in Wuhan but also the experience acquired in the interim in our hospital. Additionally, 39.8% of FMS had poor sleep quality, which was lower than the 51.7% of FMS experiencing poor sleep quality in Wuhan [9]. The high prevalence of poor sleep quality in Wuhan could be attributed to the fact that the city was the epicenter of the pandemic, hence, the high intensity and excessive pressure from rescue/relief efforts, as well as frequent/irregular shift work, could have culminated in more serious and prevalent sleep disorders for FMS.

There were several factors that may have resulted in poor sleep quality in FMS. The study demonstrated anxiety [odd ratio (OR), 4.7; 95% confidence interval (CI), 2.0, 11.2], depression (OR, 4.9; 95% CI, 2.9, 8.3), and prevalence of those who had divorced/widowed (OR, 6.1;95% CI, 1.1, 32.7) were more common in FMS with poor sleep quality than in participants with good sleep quality during the COVID-19 outbreak. COVID-19 can be transmitted via close human-to-human contact [3] and FMS are among the most vulnerable groups to infection. Therefore, anxiety and depression are also common negative emotions experienced by FMS during the

**Table 3. Comparison of the total score and subscales of sleep quality of front-line medical staff with different demographic characteristics and mental state.**

| Variables | N = 543 | PSQI score | Sleep quality | Sleep latency | The amount of sleep | Sleep efficiency | Sleep disorders | The hypnotic drug | Diurnal dysfunction |
|---|---|---|---|---|---|---|---|---|---|
| **Gender** | | | | | | | | | |
| Male | 31 | 5.20±3.45 | 0.90±0.70 | 0.65±0.66 | 0.84±0.73 | 0.58±0.57 | 0.90±0.53 | 0.09±0.19 | 1.13±0.99 |
| Female | 512 | 5.19±3.23 | 0.93±0.746 | 0.91±0.71 | 0.70±0.65 | 0.58±0.89 | 0.94±0.58 | 0.12±0.49 | 1.0±0.9 |
| t | | -0.309 | -0.207 | -2.050* | 1.147 | 0.015 | -0.334 | -1.354 | 0.747 |
| P | | 0.758 | 0.836 | 0.041 | 0.252 | 0.988 | 0.738 | 0.176 | 0.455 |
| **Age (years)** | | | | | | | | | |
| 18~34 | 324 | 4.98±0.054 | 0.90±0.747 | 0.93±0.69 | 0.62±0.66 | 0.53±0.86 | 0.94±0.61 | 0.08±0.37 | 0.99±0.86 |
| 35~44 | 130 | 5.12±0.080 | 0.96±0.720 | 0.88±0.73 | 0.72±0.57 | 0.57±0.93 | 0.92±0.54 | 0.06±0.34 | 1.02±0.93 |
| 45~54 | 77 | 6.05±0.70 | 1.06±0.71 | 0.83±0.69 | 1.04±0.61 | 0.81±1.01 | 0.97±0.53 | 0.27±0.77 | 1.06±1.01 |
| 55~64 | 12 | 5.33±0.74 | 0.67±0.98 | 0.67±0.77 | 0.75±0.96 | 0.42±0.99 | 1.00±0.60 | 0.67±1.15 | 1.17±0.93 |
| F | | 2.293 | 1.671 | 0.843 | 1.492 | 2.004 | 0.208 | 9.545*** | 0.264 |
| P | | 0.077 | 0.172 | 0.471 | 0.222 | 0.112 | 0.891 | < 0.001 | 0.852 |
| **Marital status** | | | | | | | | | |
| Married | 394 | 5.34±3.27 | 0.94±0.74 | 0.91±0.71 | 0.74±0.64 | 0.63±0.94 | 0.96±0.56 | 0.12±0.49 | 1.04±0.91 |
| Unmarried | 137 | 4.56±3.10 | 0.88±0.72 | 0.85±0.67 | 0.58±0.68 | 0.42±0.79 | 0.85±0.63 | 0.07±0.39 | 0.91±0.85 |
| Divorced or widowed | 12 | 6.75±0.76 | 1.25±0.75 | 1.00±0.85 | 1.00±0.42 | 0.75±0.75 | 1.08±0.51 | 0.25±0.86 | 1.42±0.99 |
| F | | 4.417* | 1.401 | .434 | 4.553* | 3.052* | 2.116 | 1.056 | 2.299 |
| P | | 0.013 | 0.247 | 0.648 | 0.011 | 0.048 | 0.122 | 0.348 | 0.101 |
| **Education** | | | | | | | | | |
| College degree or below | 505 | 5.18±3.27 | 0.92±0.74 | 0.90±0.71 | 0.71±0.66 | 0.60±0.92 | 0.94±0.58 | 0.12±0.50 | 0.99±0.89 |
| Bachelor's degree | 28 | 5.04±2.48 | 0.93±0.60 | 0.86±0.59 | 0.71±0.53 | 0.21±0.41 | 0.93±0.53 | 0 | 1.39±0.95 |
| Master's degree or above | 10 | 5.30±4.02 | 1.20±1.03 | 0.80±0.78 | 0.60±0.51 | 0.50±1.08 | 1.00±0.94 | 0 | 1.20±0.91 |
| F | | 0.034 | 0.672 | 0.147 | 0.136 | 2.447 | 0.061 | 1.138 | 2.922 |
| P | | 0.967 | 0.511 | 0.864 | 0.873 | 0.088 | 0.941 | 0.321 | 0.055 |
| **Living condition** | | | | | | | | | |
| Family cohabitation | 362 | 5.20±3.06 | 0.93±0.72 | 0.81±0.71 | 0.69±0.63 | 0.56±0.85 | 0.96±0.55 | 0.10±0.44 | 1.01±0.89 |
| Live alone | 181 | 5.13±3.59 | 0.93±0.78 | 0.94±0.70 | 0.73±0.70 | 0.61±1.00 | 0.88±0.63 | 0.14±0.55 | 1.01±0.93 |
| t | | 0.243 | 0.041 | 1.982 | -0.692 | -0.635 | 1.504 | -1.003 | 0 |
| P | | 0.808 | 0.967 | 0.048 | 0.489 | 0.526 | 0.133 | 0.316 | 1.000 |
| **Department** | | | | | | | | | |
| Clinical departments | 427 | 5.29±3.26 | 0.95±0.75 | 0.94±0.71 | 0.70±0.65 | 0.58±0.89 | 0.96±0.59 | 0.12±0.49 | 1.02±0.90 |
| Executive branch | 23 | 5.13±2.89 | 0.96±0.76 | 0.74±0.61 | 0.78±0.73 | 0.43±0.84 | 0.91±0.41 | 0 | 1.30±0.97 |
| Logistics department | 93 | 4.67±3.21 | 0.82±0.69 | 0.74±0.64 | 0.71±0.63 | 0.59±.99 | 0.82±0.57 | 0.11±0.47 | 0.88±0.89 |
| F | | 1.403 | 1.295 | 3.611* | 0.162 | 0.303 | 2.462 | 0.701 | 2.207 |
| P | | 0.247 | 0.275 | 0.028 | 0.851 | 0.739 | 0.086 | 0.497 | 0.111 |
| **Profession** | | | | | | | | | |
| Doctor | 43 | 5.23±3.10 | 0.88±0.69 | 0.81±0.62 | 0.70±0.55 | 0.51±0.91 | 1.02±0.59 | 0.07±0.33 | 1.23±0.99 |
| Nurse | 458 | 5.14±3.27 | 0.92±0.76 | 0.91±0.71 | 0.69±0.65 | 0.58±0.90 | 0.92±0.60 | 0.12±0.50 | 0.99±0.88 |
| Technician | 42 | 5.50±3.08 | 1.05±0.58 | 0.83±0.73 | 0.90±0.75 | 0.60±0.93 | 1.00±0.38 | 0.07±0.34 | 1.05±1.01 |
| F | | 0.243 | 0.626 | 0.551 | 2.065 | 0.129 | 0.829 | 0.408 | 1.488 |
| P | | 0.784 | 0.535 | 0.577 | 0.128 | 0.879 | 0.437 | 0.665 | 0.227 |
| **Anxiety** | | | | | | | | | |
| Yes | 43 | 8.74±3.88 | 1.56±0.88 | 1.44±0.66 | 1.09±0.75 | 0.86±1.10 | 1.56±0.79 | 0.30±0.86 | 1.93±0.91 |

(*Continued*)

**Table 3.** (Continued)

| Variables | N = 543 | PSQI score | Sleep quality | Sleep latency | The amount of sleep | Sleep efficiency | Sleep disorders | The hypnotic drug | Diurnal dysfunction |
|---|---|---|---|---|---|---|---|---|---|
| No | 500 | 4.87±2.99 | 0.88±0.70 | 0.85±0.69 | 0.67±0.63 | 0.55±0.88 | 0.88±0.53 | 0.10±0.43 | 0.93±0.86 |
| *t* | | | 7.929*** | 5.957*** | 5.410*** | 4.066*** | 2.132*** | 7.615*** | 2.672*** | 7.264*** |
| *P* | | | < 0.001 | < 0.001 | < 0.001 | < 0.001 | < 0.001 | < 0.001 | < 0.001 | < 0.001 |
| **Depression** | | | | | | | | | | |
| Yes | 103 | 7.96±3.39 | 1.44±.77 | 1.29±0.73 | 0.96±0.72 | 0.88±1.09 | 1.36±0.60 | 0.23±0.70 | 1.80±0.80 |
| No | 440 | 4.52±2.84 | 0.81±0.68 | 0.80±0.66 | 0.65±0.62 | 0.51±0.84 | 0.84±0.53 | 0.09±0.41 | 0.83±0.82 |
| *t* | | | 10.626*** | 8.139*** | 6.535*** | 4.428*** | 3.840*** | 8.658*** | 2.786*** | 10.767*** |
| *P* | | | < 0.001 | < 0.001 | < 0.001 | < 0.001 | < 0.001 | < 0.001 | < 0.001 | < 0.001 |

*0.05 > p-value > = 0.01.

**0.01 > p-value > = 0.001.

***p-value < 0.001.

COVID-19 outbreak [26]. Negative emotions and repetitive negative thinking patterns are associated with problems in initiating and maintaining sleep [27]. Our results showed that compared with the FMS without anxiety or depression, in FMS with either anxiety or depression, the duration of sleep decreased, sleep latency increased, sleep efficiency decreased, and wake-up time increased. Sleep disorders and diurnal dysfunction are serious concerns during the COVID-19 outbreak, thus, leading to poor sleep quality. Marital status is also a risk factor for poor sleep quality in FMS during the COVID-19 outbreak. The poor sleep quality of divorced/widowed FMS is mainly reflected in the lack or low efficiency of sleep during the COVID-19 outbreak, possibly due to the fact that divorced/widowed FMS may lack certain social support (peer communication and emotional support) during the outbreak, which indirectly leads to the decline of sleep quality [28]. The study also found that in terms of sleep latency scores, the values were higher in female FMS than in their male counterparts, and FMS living alone had higher scores those living with their families. In China, at the end of a paid professional work shift, women are generally expected to take on additional family responsibilities. The fact that women are expected to simultaneously assume multiple social roles invariably exerts a disproportionate pressure on women, which may lead to increased sleep latency. We also found that the hypnotic drug score of FMS aged 55–64 years was higher than that of other age groups. This suggests that older FMS are more likely to consume hypnotic drugs to improve the quality of sleep.

During the epidemic period, a large number of medical staff were sent to Wuhan for support, resulting in a shortage of medical staff in our hospital. As a consequence, the clinical

**Table 4. Analysis of variance between different groups, endpoint: PSQI.**

| Variables | N (%) | M±SD | F | P-value |
|---|---|---|---|---|
| Anxiety only(group1) | 16 (2.9) | 7.58±3.15 | 50.8*** | < 0.001 |
| Depression only (group 2) | 76(14.0) | 8.25±4.01 | | |
| Both anxiety and depression (group 3) | 27(5.0) | 9.04±3.85 | | |
| Neither anxiety nor depression (group4) | 424(78.1) | 4.38±2.69 | | |

The statistical methods used for comparisons were the One-Way Anova (normal distribution).

M = Mean.

SD = Standard difference.

***p-value < 0.001.

**Table 5. Repeated measures of each dependent variable.**

| Comparison group | MD | SE | *P-value* | *95%CI* |
|---|---|---|---|---|
| 1and 2 | -0.671 | 0.791 | 0.396 | -2.22–0.88 |
| 2and 3 | -1.458 | 0.644 | 0.024 | -2.72–-0.19 |
| 2and 4 | 3.197 | 0.358 | 0.000 | 2.49–3.90 |
| 1and 3 | -0.787 | 0.907 | 0.386 | -2.57–0.99 |
| 1and 4 | 3.868 | 0.732 | 0.000 | 2.43–5.31 |
| 3and 4 | 4.655 | 0.571 | 0.000 | 3.53–5.78 |

MD = Mean difference.

SE = Standard Error.

CI = confidence interval.

medical staff needed to alternate their shifts frequently to accommodate the busy clinical work. Medical staff on duty must always be on active standby. If they are repeatedly awakened at night, the steady state of sleep/wake cycle will be interrupted, and the sleep state will be fragmented, which will make it difficult to fall asleep again and lead to sleep disorder. When the medical staff on shift attempt to supplement sleep during daytime, they are invariably exposed to strong natural light and a noisy rest environment during the day, which is conducive to premature and spontaneous sleep termination during the sleep period that ensues the night shift. As a result, the overall sleep duration is significantly shortened, resulting in an overall decline in sleep quality. Persistent work-related pressure, frequent sleep deprivation and irregular shift work, collectively precipitate sleep disorders and sleep schedule disorders, which ultimately lead to a decline in overall sleep quality.

We analyzed the relationship between mental state and sleep quality. The results suggest that the sleep quality of the FMS experiencing only anxiety, only depression, or co-occurrence of anxiety and depression was worse than those with neither anxiety nor depression during the COVID-19 outbreak. In addition, the results indicate that the sleep quality of FMS with co-occurrence of anxiety and depression was worse than those with only depression. Depression can lead to poor sleep quality, but the overlapping or potentially cumulative effects of depression and anxiety can exacerbate the decline in sleep quality. A study also concluded that patients with both depression and anxiety symptoms have a higher incidence of sleep disorders [29]. However, in our study, the sleep quality of FMS with anxiety was not much different than the sleep quality of FMS with both anxiety and depression. It is plausible that patients with anxiety are prone to experiencing serious sleep disorders, because sleep disorders and anxiety have a common/underlying pathogenesis: hyperactivity caused by disorders of neurotransmitter systems, such as cholinergic and GABA [30].

Sleep disorders could be an early symptom, part of a prodrome, of an underlying depressive or anxiety disorder. Similarly, sleep disorders might also exist as a separate, comorbid disorder that either gave rise to or developed from an undiagnosed psychiatric condition. Furthermore, anxiety and depression have bidirectional association with sleep quality [31]. Continuous poor sleep quality will lead to decreased daytime function, emotional instability, and mental exhaustion, thus increasing the risk of depression and anxiety [32]. Additionally, it is known that anxiety, depression and sleep disorders intersect by mutually affecting and triggering/exacerbating each other. The fundamental underpinning of insomnia is through to be cognition in the form of anxiety [33]. Depression can predispose individuals to unreasonable beliefs about sleep disorders, leading to a more serious anxiety state. Insomnia forms an emotional memory in the anxiety and depression, conditionally activating sympathetic nervous system, further

aggravating the existing state of anxiety, and becoming a self-sustaining malignant cycle, which keeps individuals in a highly awake state, leading to persistent sleep disorders. Particularly during the COVID-19 outbreak, FMS faced greater risk of infection and work stress, as well as frequent policy changes, unclear case management criteria, and other ambiguous conditions that led to depression and anxiety, which have contributed to the higher incidence of sleep disorders.

Therefore, under the stress of an epidemic outbreak, hospitals must be vigilant and proactive in assessing the mental health and sleep quality of FMS as well as extending remediation efforts to help them cope with existing mental health/sleep quality issues. During clinical intervention of FMS who experience psychological problems and sleep disorders, the clinicians should look for the potential comorbidity mental state, so as to determine the appropriate treatment modality to break the vicious cycle of anxiety, depression and sleep disorders, thereby, improving the mental health and sleep quality.

For most FMS, what they need is more undisturbed rest [34]. We advocated that hospitals should establish a shift system to allow their FMS to rest and take turns to undertake high-risk and high-pressure work. A detailed plan in advance may improve the effectiveness of post-disaster interventions, such as effective risk communication and the provision of psychological first aid [35]. Through effective training and support, hospitals can provide online consultation platforms, disseminate information on how to reduce the risk of transmission amongst FMS in a clinical setting, and provide timely and authoritative information on pandemic dynamics, which may be instrumental in reducing the psychological impact on FMS. In addition, a tight social support network can help medical staff reduce or better manage their anxiety levels, thus, indirectly help to improve sleep quality. Therefore, it is suggested that family members or close friends provide a compassionate and supportive social network for FMS as a form of emotional and social support. Also, hospitals and governments should provide welfare subsidies to FMS in order to alleviate financial burdens on them during these extraordinary times.

When FMS experience a decline in sleep quality or insomnia, it is advisable that a number of rectifying measures be adopted to improve sleep quality, including sleep health education, relaxation training, and cognitive behavioral therapy to conduct self-regulation (Cognitive behavioral therapy for insomnia, CBT-I). As a final resort, if the aforementioned measures are ineffective, hypnotics, as psychoactive compounds, can be administered under the guidance of psychiatrists. It should also be underscored that excessive publicity or over-reporting of the adverse circumstances of FMS fighting the COVID-19 can do a disservice to them by demoralizing them, which may lead to symptoms such as over-excitement, irritability, psychological distress, unwillingness to take sufficient rest, etc. It is, therefore, our suggestion that media outlets and news agencies avoid excessive reporting and coverage of the circumstances of the FMS fighting the epidemic.

## Limitations

This investigation has several limitations that need to be considered in the interpretation of our results. First, this is a cross-sectional study. As such, we cannot infer causality in the interpretation of the findings. Second, our survey subjects were derived from one specific hospital through convenience sampling, and the proportion of male subjects in this study was low, thus the representativeness may be reduced. Third, the participants completed the self-reported questionnaires using the WeChat application and mobile devices, which might lead to self-selection bias. Nevertheless, assessing anxiety and depressive symptoms via an individualized interview with a qualified psychiatrist would have been ideal. Moreover, the causality or subordination between anxiety, depression and poor sleep quality is not self-evident. Therefore, the

relationship between anxiety, depression, and sleep quality of medical staff in public health emergencies may have other directions, which can be explored in future studies.

## Conclusions

An observational and cross-sectional clinical study was conducted to investigate the mental state and sleep quality of the FMS during the COVID-19 outbreak in a hospital setting. The results suggested that anxiety, depression, and divorce/bereavement were more common in the FMS with poor sleep quality than in those with good sleep quality. The sleep quality of FMS with co-occurrence of anxiety and depression was worse than that of the medical staff with depression alone. Therefore, during the epidemic period, particular attention must be paid to the mental well-being and sleep quality of FMS. Strategies aimed at prevention and timely intervention of sleep disorders, anxiety and depression in FMS are crucial to help us effectively treat and contain the recent pandemic in hospital settings.

## Supporting information

**S1 File. The data used and/or analyzed during the current study.**
(XLS)

## Acknowledgments

The authors thank the medical staff who participated in the study.

## Author Contributions

**Conceptualization:** Jianying Yu, Xiandong Meng.

**Data curation:** Yu Zhuo.

**Formal analysis:** Leiyu Yue, Rui Zhao.

**Funding acquisition:** Jianying Yu.

**Investigation:** Leiyu Yue, Rui Zhao, Qingqing Xiao.

**Methodology:** Leiyu Yue, Qingqing Xiao.

**Project administration:** Xiandong Meng.

**Resources:** Rui Zhao, Jianying Yu.

**Software:** Yu Zhuo.

**Supervision:** Xiandong Meng.

**Validation:** Jianying Yu.

**Visualization:** Yu Zhuo.

**Writing – original draft:** Leiyu Yue.

**Writing – review & editing:** Xiandong Meng.

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
