## [Decision Letter · Decision Letter 0]

1 Feb 2021

PONE-D-20-30673

The Effect of mental health on Sleep Quality of front-line medical staff during the COVID-19 outbreak in China: A cross-sectional study

PLOS ONE

Dear Dr. Meng,

Thank you for submitting your manuscript to PLOS ONE. After careful consideration, we feel that it has merit but does not fully meet PLOS ONE’s publication criteria as it currently stands. Therefore, we invite you to submit a revised version of the manuscript that addresses the points raised during the review process.

We look forward to receiving your revised manuscript.

Kind regards,

Stephan Doering, M.D.

Academic Editor

PLOS ONE

2. In your Methods section, please provide additional information about the participant recruitment method and the demographic details of your participants. Please ensure you have provided sufficient details to replicate the analyses such as:   

-    a description of any inclusion/exclusion criteria that were applied to participant recruitment,

-    a statement as to whether your sample can be considered representative of a larger population,

-    a description of how participants were recruited, and

-       descriptions of where participants were recruited and where the research took place

-       a sample size calculation.

" The funders had no role in study design, data collection and analysis, decision to publish, or preparation of the manuscript."

Reviewers' comments:

Reviewer's Responses to Questions

**Comments to the Author**

1. Is the manuscript technically sound, and do the data support the conclusions?

Reviewer #1: Yes

Reviewer #2: Yes

2. Has the statistical analysis been performed appropriately and rigorously? 

Reviewer #1: Yes

Reviewer #2: No

3. Have the authors made all data underlying the findings in their manuscript fully available?

Reviewer #1: No

Reviewer #2: Yes

4. Is the manuscript presented in an intelligible fashion and written in standard English?

Reviewer #1: Yes

Reviewer #2: Yes

5. Review Comments to the Author

Reviewer #1: The Current manuscript addressing the topic “The Effect of mental health on Sleep Quality of front-line medical staff during the COVID-19 outbreak in China: A cross-sectional study” is a good effort to explore the mental health and quality of sleep of front-line medical staff during pandemic like COVID-19. Overall the article is well written but I have a few suggestions and comments,

Abstract

Kindly improve the presentation of the structured abstract. In the presentation of results, it is suggested to use only P-values instead of OR, CI.

Introduction

It is suggested to include the data of the outbreak during the period when the study was conducted to provide more context to interpret the findings.

Methods

The methodology of the study is not described properly.

Is the study conducted physically or by using an online platform to administer the questionnaire?

Is there any strategy used to address the missing data?

There is no description, how the sample size was calculated, and how the participants were recruited to achieve the final number of participants for analysis?

There are no details provided on what basis multivariate logistic regression applied to different variables.

Results

It is advisable to adopt a dual approach to report cut-offs of BDI, SAS, and PSQI with its continuous score in table 1.

Kindly clarify what test applied to analyze the factors affecting sleep quality is it the Chi-square test or logistic regression analysis also revise p-values presented in table 2.

Discussion.

To improve the discussion section and as well as the introduction section it is suggested to provide a link between the effect of COVID-19 on the mental health & sleep quality of healthcare workers.

I would suggest providing references to similar studies conducted during the current pandemic in different countries, like one recently published study “Arshad et al. Assessing the Impact of COVID-19 on the Mental Health of Healthcare Workers in Three Metropolitan Cities of Pakistan. Psychology research and behavior management, 13, 1047. Doi.10.2147/PRBM.S282069.

Limitation

Is there any strategy used to remove biases, if not mention it as a limitation to the current study.

Reviewer #2: Estimated Authors,

Estimated Editors,

I've appreciated your valuable paper on the Effect of mental health on Sleep Quality of front-line medical staff during the

COVID-19 outbreak in China. Even though the results of your study were largely not unexpected (i.e. we are dealing with a significant stressor event, and its consequences are both consistent with previous evidences and international daily experience of healthcare workers), it doesn't mean that such results may be of limited interest - viceversa, I'm confident that the work of Meng et al. may contribute to the global effort against the ongoing pandemic.

In my opinion, introduction, results and discussion are properly reported. My only concern refers to the sample size. Authors should explain whether their sample was collected by convenience or a preventive power analysis was otherwise modelled. It is particularly important as Authors did perform a multivariate analysis, whose reliability should be discussed accordingly to the actual reliability of statistical estimates.

6. PLOS authors have the option to publish the peer review history of their article (what does this mean?). If published, this will include your full peer review and any attached files.

Reviewer #1: No

Reviewer #2: **Yes: **Matteo Riccò

---

## [Author Response · Author response to Decision Letter 0]

11 Mar 2021

Dear editor:

Thanks a lot for your reviews to our manuscript. We acknowledge your comments and constructive suggestions very much, which are valuable for improving the quality of our manuscript. We have revised the manuscript according to the reviewers’ comments in detail. We hope, with these modifications and improvements based on your suggestions and the reviewers’ comments, the quality of our manuscript would meet the publication standard of PLOS ONE.

The revisions have been done in the attached manuscript. Some explanations regarding the revisions of our manuscript are as follows. If you have any question, please contact us without hesitate.

Academic editor

Q1：Please ensure that your manuscript meets PLOS ONE's style requirements, including those for file naming.

A1：Thank you for bringing up this important point. We have revised the format of the article as required.

Q2：In your Methods section, please provide additional information about the participant recruitment method and the demographic details of your participants. Please ensure you have provided sufficient details to replicate the analyses such as: a description of any inclusion/exclusion criteria that were applied to participant recruitment, a statement as to whether your sample can be considered representative of a larger population, a description of how participants were recruited. descriptions of where participants were recruited and where the research took place, a sample size calculation.

A2: Thank you for bringing up this important point. We have revised the Methods section and supplemented the content which was omitted before. 

 This study included 543 FMS from a medical center in Western China who regularly treated or were in contact with patients infected with COVID-19, during a period spanning from February, 2020 to March, 2020. The inclusion criteria were as follows: (1) regular employees, (2) worked at their posts during the survey, (3) WeChat users. All study participants willingly volunteered to participate in the study.

 The subjects of this study were selected from the medical staff of West China Hospital of Sichuan University，which is one of the largest medical centers in China. After the outbreak of the epidemic, we not only sent medical staff to Wuhan to treat the infected people in the first time, but also undertook the treatment of the infected people in the area where the hospital is located. Besides, in this study, 84.4% of the research population were nurses, and 94.3% of the research population were women, which was consistent with the occupation and sex distribution of the Chinese healthcare system. However, our survey subjects were derived from one specific hospital through convenience sampling, thus the representativeness may be reduced, which may be treated as a limitation of the study.

 We contacted department heads in each department and invited them to forward our questionnaire to their WeChat group of staff to recruit participants.

 The questionnaire was built on a professional questionnaire survey network platform called“Wenjuan Xing” (www.wjx.cn) and then was shared on social media WeChat.

 Sample size was determined using the formula:

N=(z^2×p(1-p))/e^2 

where ‘z’ is 1.96 at 95% confidence interval, ‘e’ is margin of error at 5% and ‘p’ is prevalence rate of 40% from a recent study done in China.

Q3: About financial disclosure: please address the following queries: 

 Please clarify the sources of funding (financial or material support) for your study. List the grants or organizations that supported your study, including funding received from your institution.

 State what role the funders took in the study. If the funders had no role in your study, please state: “The funders had no role in study design, data collection and analysis, decision to publish, or preparation of the manuscript.”

 If any authors received a salary from any of your funders, please state which authors and which funders.

A3: This research was received grant from funding 2019 novel coronavirus disease technology research project of West China Hospital of Sichuan University (NO: HX-2019-nCov-034). The funders had no role in study design, data collection and analysis, decision to publish, or preparation of the manuscript. None of the authors received a salary from the funders.

Reviewer #1

Abstract: Kindly improve the presentation of the structured abstract. In the presentation of results, it is suggested to use only P-values instead of OR, CI.

A: Thanks for the detailed suggestion. We have modified it according to your suggestion.

Introduction: It is suggested to include the data of the outbreak during the period when the study was conducted to provide more context to interpret the findings.

A: Thanks for the detailed suggestion. We supplemented studies on the incidence of anxiety, depression, and sleep disorders among health care workers during the epidemic. such as，in a meta-analysis showed that anxiety was assessed in 12 studies, with a pooled prevalence of 23.2% and depression in 10 studies, with a prevalence rate of 22.8% during the COVID-19 pandemic. Qiu et al found that 39.2% of Chinese medical workers suffered from sleep disorders. However, 51.7% of front-line medical staff had sleep disorders under COVID-19 outbreak in China.

Methods: 

Thank you for your insightful advisement and comments on the Methods section. All your questions are answered below.

Q1: Is the study conducted physically or by using an online platform to administer the questionnaire?

A1: The questionnaire was built on a professional questionnaire survey network platform called “Wenjuan Xing” (www.wjx.cn) and then was shared on social media WeChat.

Q2: Is there any strategy used to address the missing data?

A2: While constructing the online questionnaire, the integrity check function of the platform was used, meaning the questionnaire could not be submitted unless all questions were answered.

Q3: There is no description, how the sample size was calculated, and how the participants were recruited to achieve the final number of participants for analysis?

A3: Sample size was determined using the formula:

N=(z^2×p(1-p))/e^2 

where ‘z’ is 1.96 at 95% confidence interval, ‘e’ is margin of error at 5% and ‘p’ is prevalence rate of 40% from a recent study done in China. According to the formula, N=369, considering the non-response rate of 20%, at least 443 sample size are needed. We contacted department heads in each department and invited them to forward our questionnaire to their WeChat group of staff to recruit participants. A total of 546 FMS completed the questionnaire survey, which included three participants were disagreeing to use their questionnaire for study due to their answers are worthless(n=2), worried about expose their privacy(n=1). Hence,543 effective questionnaires for analysis.

Q4: There are no details provided on what basis multivariate logistic regression applied to different variables.

A4: Multivariate logistic regression analysis was performed on the variables that were significant in univariate analysis.

Results:

Q1: It is advisable to adopt a dual approach to report cut-offs of BDI, SAS, and PSQI with its continuous score in table 1.

A1：Thanks for the detailed suggestion. We have made modifications according to your suggestions and the format of the whole table has been revised.

Q2: Kindly clarify what test applied to analyze the factors affecting sleep quality is it the Chi-square test or logistic regression analysis also revise p-values presented in table 2.

A2: The binary logistic regression analyses were used to estimate the odds ratio for each independent variable, to assess which of the factors associated with poor sleep quality. In addition, p-values presented in table 2 have been revised.

Discussion:

Q: To improve the discussion section and as well as the introduction section it is suggested to provide a link between the effect of COVID-19 on the mental health & sleep quality of healthcare workers.

A: Thank you for pointing it out. In fact, anxiety and depression have bidirectional association with sleep quality. Continuous poor sleep quality will lead to decreased daytime function, emotional instability, and mental exhaustion, thus increasing the risk of depression and anxiety. Additionally, it is known that anxiety, depression and sleep disorders intersect by mutually affecting and triggering/exacerbating each other. Particularly during the COVID-19 outbreak, FMS faced greater risk of infection and work stress, as well as frequent policy changes, unclear case management criteria, and other ambiguous conditions that led to depression and anxiety. Insomnia forms an emotional memory in the anxiety and depression, conditionally activating sympathetic nervous system, further aggravating the existing state of anxiety, and becoming a self-sustaining malignant cycle, which keeps individuals in a highly awake state, leading to persistent sleep disorders.

Limitation:

Q: Is there any strategy used to remove biases, if not mention it as a limitation to the current study.

A: Thank you for bringing up this important point. The participants completed the self-reported questionnaires using the WeChat application and mobile devices, which might lead to self-selection bias. Therefore, it will be a limitation of this study.

Reviewer #2

Q: In my opinion, introduction, results and discussion are properly reported. My only concern refers to the sample size. Authors should explain whether their sample was collected by convenience or a preventive power analysis was otherwise modelled. It is particularly important as Authors did perform a multivariate analysis, whose reliability should be discussed accordingly to the actual reliability of statistical estimates.

A: Thank you very much for your very important suggestions and comments on the sample size. In fact, a convenience sampling method was used to recruit participants in this study. According to inclusion and exclusion criteria，a total of 546 FMS completed the questionnaire survey, which included three participants were disagreeing to use their questionnaire for study due to their answers are worthless(n=2), worried about expose their privacy(n=1), 543 effective questionnaires for analysis. Based on the calculation method of cross-sectional study sample size, at least 443 sample size are needed for this study. Hence, the participants we recruited reached the estimated sample size.

Sample size was determined using the formula:

N=(z^2×p(1-p))/e^2 

where ‘z’ is 1.96 at 95% confidence interval, ‘e’ is margin of error at 5% and ‘p’is prevalence rate of 40% from a recent study done in China. According to the formula, N=369, considering the non-response rate of 20%, at least 443 sample size are needed.

---

## [Decision Letter · Decision Letter 1]

9 Apr 2021

PONE-D-20-30673R1

The Effect of mental health on Sleep Quality of front-line medical staff during the COVID-19 outbreak in China: A cross-sectional study

PLOS ONE

Dear Dr. Meng,

Thank you for submitting your manuscript to PLOS ONE. After careful consideration, we feel that it has merit but does not fully meet PLOS ONE’s publication criteria as it currently stands. The reviewers basically are in favor of your manuscript, however, Reviewer two raises one important issue. May I ask you to take care of that? Therefore, we invite you to submit a revised version of the manuscript that addresses the points raised during the review process.

We look forward to receiving your revised manuscript.

Kind regards,

Stephan Doering, M.D.

Academic Editor

PLOS ONE

Journal Requirements:

Additional Editor Comments (if provided):

Reviewers' comments:

Reviewer's Responses to Questions

**Comments to the Author**

1. If the authors have adequately addressed your comments raised in a previous round of review and you feel that this manuscript is now acceptable for publication, you may indicate that here to bypass the “Comments to the Author” section, enter your conflict of interest statement in the “Confidential to Editor” section, and submit your "Accept" recommendation.

Reviewer #1: All comments have been addressed

Reviewer #2: (No Response)

2. Is the manuscript technically sound, and do the data support the conclusions?

Reviewer #1: Yes

Reviewer #2: Yes

3. Has the statistical analysis been performed appropriately and rigorously? 

Reviewer #1: Yes

Reviewer #2: Yes

4. Have the authors made all data underlying the findings in their manuscript fully available?

Reviewer #1: Yes

Reviewer #2: Yes

5. Is the manuscript presented in an intelligible fashion and written in standard English?

Reviewer #1: Yes

Reviewer #2: Yes

6. Review Comments to the Author

Reviewer #1: The authors have adequately answered all of my queries in their revised submission and I hope that if this manuscript is accepted it will definitely get attention of the readers.

Reviewer #2: Estimated Authors,

I've read with interest your revised paper. The main text has been largely amended in accordance with my previous recommendations. However, a minor remark regarding the section on the sample size calculation:

you wrote:

Sample size was determined using the formula:

N=(z^2×p(1-p))/e^2

where ‘z’ is 1.96 at 95% confidence interval, ‘e’ is margin of error at 5% and ‘p’is

prevalence rate of 40% from a recent study done in China. According to the formula,

N=369,

It is rather unclear what the prevalence rate of 0.4 refers to. Please explain.

7. PLOS authors have the option to publish the peer review history of their article (what does this mean?). If published, this will include your full peer review and any attached files.

Reviewer #1: No

Reviewer #2: **Yes: **Matteo Riccò

---

## [Author Response · Author response to Decision Letter 1]

5 May 2021

Academic editor

Q：Please review your reference list to ensure that it is complete and correct.

A: Thank you for bringing up this important point. We have revised the format of the references as required. In addition, two references were replaced, the 33rd and 35th respectively. The 33rd is the research published in Chinese journals, and the 35th is the operation manual. Readers may not get the full text, so we use the relevant references instead. 

Reviewer#2

Q: Sample size was determined using the formula:

N=(z^2×p(1-p))/e^2 

where ‘z’ is 1.96 at 95% confidence interval, ‘e’ is margin of error at 5% and ‘p’ is prevalence rate of 40% from a recent study done in China. According to the formula, N=369, It is rather unclear what the prevalence rate of 0.4 refers to. Please explain.

A: Thank you for pointing it out. In fact, the aim of this study was to investigate the sleep quality of FMS during the COVID-19 outbreak in China and analyze the relationship between mental health and sleep quality of FMS. The logistic regression analyses were used to estimate the odds ratio for each independent variable, to assess which of the factors associated with sleep disorders. Therefore, the prevalence rate of 0.4 refers to the prevalence rate of sleep disorders in Chinese healthcare professionals under the outbreak of COVID-19.

---

## [Decision Letter · Decision Letter 2]

14 Jun 2021

The Effect of mental health on Sleep Quality of front-line medical staff during the COVID-19 outbreak in China: A cross-sectional study

PONE-D-20-30673R2

Dear Dr. Meng,

We’re pleased to inform you that your manuscript has been judged scientifically suitable for publication and will be formally accepted for publication once it meets all outstanding technical requirements.

Kind regards,

Stephan Doering, M.D.

Academic Editor

PLOS ONE

Reviewer's Responses to Questions

**Comments to the Author**

1. If the authors have adequately addressed your comments raised in a previous round of review and you feel that this manuscript is now acceptable for publication, you may indicate that here to bypass the “Comments to the Author” section, enter your conflict of interest statement in the “Confidential to Editor” section, and submit your "Accept" recommendation.

Reviewer #2: All comments have been addressed

2. Is the manuscript technically sound, and do the data support the conclusions?

Reviewer #2: Yes

3. Has the statistical analysis been performed appropriately and rigorously? 

Reviewer #2: Yes

4. Have the authors made all data underlying the findings in their manuscript fully available?

Reviewer #2: Yes

5. Is the manuscript presented in an intelligible fashion and written in standard English?

Reviewer #2: Yes

6. Review Comments to the Author

Reviewer #2: All my concerns have been addressed. Therefore, I've no further recommendations to the study Authors, and I endorse the final publication of this paper.

7. PLOS authors have the option to publish the peer review history of their article (what does this mean?). If published, this will include your full peer review and any attached files.

Reviewer #2: **Yes: **Matteo Riccò

---

## [Editor Report · Acceptance letter]

16 Jun 2021

PONE-D-20-30673R2 

The effect of mental health on sleep quality of front-line medical staff during the COVID-19 outbreak in China: A cross-sectional study 

Dear Dr. Meng:

I'm pleased to inform you that your manuscript has been deemed suitable for publication in PLOS ONE. Congratulations! Your manuscript is now with our production department. 

Kind regards, 

on behalf of

Professor Stephan Doering 

Academic Editor

PLOS ONE